# Diagnosis and Antifungal Prophylaxis for COVID-19 Associated Pulmonary Aspergillosis

**DOI:** 10.3390/antibiotics11121704

**Published:** 2022-11-26

**Authors:** Armani M. Hawes, Nitipong Permpalung

**Affiliations:** Department of Medicine, Johns Hopkins University School of Medicine, Baltimore, MD 21205, USA

**Keywords:** CAPA, fungal prophylaxis, COVID-19, aspergillosis

## Abstract

The COVID-19 pandemic has redemonstrated the importance of the fungal-after-viral phenomenon, and the question of whether prophylaxis should be used to prevent COVID-19-associated pulmonary aspergillosis (CAPA). A distinct pathophysiology from invasive pulmonary aspergillosis (IPA), CAPA has an incidence that ranges from 5% to 30%, with significant mortality. The aim of this work was to describe the current diagnostic landscape of CAPA and review the existing literature on antifungal prophylaxis. A variety of definitions for CAPA have been described in the literature and the performance of the diagnostic tests for CAPA is limited, making diagnosis a challenge. There are only six studies that have investigated antifungal prophylaxis for CAPA. The two studied drugs have been posaconazole, either a liquid formulation via an oral gastric tube or an intravenous formulation, and inhaled amphotericin. While some studies have revealed promising results, they are limited by small sample sizes and bias inherent to retrospective studies. Additionally, as the COVID-19 pandemic changes and we see fewer intubated and critically ill patients, it will be more important to recognize these fungal-after-viral complications among non-critically ill, immunocompromised patients. Randomized controlled trials are needed to better understand the role of antifungal prophylaxis.

## 1. Background

The coronavirus disease 2019 (COVID-19) pandemic has led to major shifts in how infectious diseases are viewed and managed, and we have seen its influence on a variety of healthcare-associated infections as well as secondary bacterial and fungal infections [1,2,3]. COVID-19-associated pulmonary aspergillosis (CAPA) is defined as pulmonary and airway aspergillosis that is diagnosed in temporal proximity to a preceding SARS-CoV-2 infection. Risk factors for CAPA include older age, underlying chronic lung disease, chronic renal failure, cancer, multiple myeloma, solid organ transplant, receiving corticosteroids, and receiving tocilizumab [4,5,6,7,8]. Intensive care unit (ICU) cohort studies have reported the incidence of CAPA to range from 5% to 30% in patients with severe COVID-19, with significant mortality [9,10,11,12,13]. The fungal-after-viral phenomenon has been previously studied with influenza-associated pulmonary aspergillosis (IAPA) [14]. Rates of IAPA range from 7–30% among people with severe influenza requiring ICU admission [15,16]. The pathophysiology of CAPA, however, is distinct from invasive aspergillosis in classically immunocompromised patients, as angioinvasion is less likely to occur, thus limiting the performance of serum fungal biomarkers [7]. Theories on the predisposition these patients have to aspergillosis include viral-mediated damage to the airway that results in a loss of ability to clear fungi and suppression of secondary immunologic defenses [17,18]. There may be a multi-level breach in immunity that predisposes patients with viral pneumonia to aspergillosis [19]. This involves a breach in the epithelium, inadequate phagocytosis, and failure of neutrophils [20,21,22]. This pathophysiology seems unique to virally mediated damage to the lungs. Moreover, COVID-19-specific treatments, in conjunction with the viral-mediated damage, may predispose patients even further to co-infections [23,24].

The challenge in the diagnosis of CAPA is that a variety of definitions have been used in the existing literature [6]. The European Organization for Research and Treatment of Cancer and the Mycoses Study Group Education and Research Consortium (EORTC/MSGERC) criteria that are typically used to diagnose invasive aspergillosis rely on host, clinical, and mycological factors which then categorize patients into proven, probable, or possible aspergillosis [25]. However, COVID-19 patients often do not meet the criteria for host factor because they are typically not immunocompromised, and they often lack additional clinical factors such as classic computed tomography (CT) findings [6]. Additionally, because they are less likely to experience angioinvasion leading to antigenemia, serum fungal biomarkers are less sensitive [7,26]. In 2020, a consensus case definition classifying patients into proven, probable, and possible CAPA was published by the European Confederation for Medical Mycology and the International Society for Human and Animal Mycology (ECMM/ISHAM) [6]. However, these definitions are highly reliant on bronchoscopy, which is not always available, as well as serum fungal biomarkers that lack sensitivity in patients without angioinvasion [7]. Additionally, fungal biomarker cutoffs may not be aligned with local regulatory recommendations and various cutoffs for non-bronchoalveolar lavage (BAL) have not been validated [27]. Because of the complexities in defining CAPA, there are uncertainties regarding its treatment and the need for prophylaxis. Hence, the aim of this work is to explore the major insights into diagnosis and antifungal prophylaxis for CAPA.

## 2. Case Definitions and Diagnosis

### 2.1. Case Definitions 

#### 2.1.1. EORTC/MSGERC

As discussed above, the classic EORTC/MSGERC definitions of possible, probable, and proven invasive aspergillosis may not be the best to capture people with CAPA [25,28]. Proven invasive pulmonary aspergillosis (IPA) requires histopathologic or microscopic evidence of hyphae and lung tissue damage. Probable diagnosis requires host factors such as neutropenia, hematologic malignancy, and solid organ transplant, as well as clinical factors, including radiographic signs consistent with invasive fungal infection or endobronchial lesions are seen via a bronchoscopy, and mycological factors such as a positive galactomannan (GM) test, polymerase chain reaction (PCR), or culture from non-sterile sites. Hence, patients with CAPA may not have classic host and clinical factors consistent with probable IPA according to the EORTC/MSGERC definitions (Figure 1).

#### 2.1.2. AspICU

As clinicians began to see more fungal-after-viral phenomena, further guidelines and definitions were created. The AspICU definitions were created to better capture IAPA cases as patients with IAPA may not have classic immunocompromised conditions according to the EORTC/MSGERC consensus definitions [29]. The authors defined proven cases as meeting the EORTC/MSGERC proven criteria. Their putative cases required a positive lower respiratory tract fungal culture, clinical signs and symptoms of disease, abnormal chest radiology, and either one host risk factor or semiquantitative *Aspergillus*-positive culture of BAL fluid without bacterial growth together with a positive cytological smear showing branching hyphae (Figure 1).

#### 2.1.3. Expert Case Definitions for IAPA

Subsequently, a group of experts proposed a case definition of IAPA through a process of informal consensus [30]. The authors used evidence of *Aspergillus* tracheobronchitis as a clinical branch point for their definition. In patients with *Aspergillus* tracheobronchitis, the authors defined proven IAPA as a biopsy or brush specimen of an airway plaque, pseudomembrane, or ulcer showing hyphal elements and *Aspergillus* growth on culture or positive *Aspergillus* PCR in tissue. The authors defined probable IAPA in patients with tracheobronchitis if patients had an airway plaque, pseudomembrane, or ulcer as well as one of the following: serum GM index > 0.5, BAL GM ≥ 1, positive BAL culture, positive tracheal aspirate culture, positive sputum culture, or hyphae consistent with *Aspergillus*. In patients without documented *Aspergillus* tracheobronchitis, the authors defined proven IAPA as a lung biopsy showing invasive fungal elements and *Aspergillus* growth on culture or positive *Aspergillus* PCR in tissue. The authors defined probable IAPA in patients without *Aspergillus* tracheobronchitis as pulmonary infiltrate with at least one of the following: serum GM index > 0.5, BAL GM ≥ 1 or positive BAL culture, or a cavitating infiltrate with a positive sputum culture or positive tracheal aspirate culture (Figure 1). The later-devised CAPA definitions, described below, were somewhat influenced by these case definitions.

#### 2.1.4. ECCM/ISHAM

The ECMM/ISHAM defines proven CAPA as pulmonary or tracheobronchial infection with *Aspergillus* that is proven by histopathologic or direct microscopic detection of fungal elements. All proven, probable, and possible definitions require an entry criterion of a diagnosis of COVID-19 severe enough to require ICU admission. The authors further define proven CAPA as the entry criterion and at least one of the following: histopathological or direct microscopic detection of fungal hyphae that shows invasive growth with associated tissue damage; or *Aspergillus* recovered by culture or microscopy or histology or PCR obtained by a sterile aspiration or biopsy from a pulmonary site, showing an infectious disease process. Probable CAPA is defined as the entry criterion as well as tracheobronchitis, indicated by tracheobronchial ulceration, nodule, pseudomembrane, plaque, or eschar with at least one of the following: microscopic detection of fungal elements in BAL; positive BAL culture or PCR; serum GM index > 0.5 or serum lateral flow assay (LFA) index > 0.5; or BAL GM index ≥ 1.0 or BAL LFA index ≥ 1.0. Possible CAPA is defined as the entry criterion, as well as a pulmonary infiltrate not attributed to another cause in combination with at least one of the following: microscopic detection of fungal elements in non-bronchoscopic lavage; positive non-bronchoscopic lavage culture; single non-bronchoscopic lavage GM index > 4.5; non-bronchoscopic lavage GM index > 1.2 twice or more, or non-bronchoscopic lavage GM index > 1.2 plus another non-bronchoscopic lavage mycology test positive (non-bronchoscopic lavage PCR or LFA) (Figure 1) [6].

### 2.2. Current CAPA Diagnostic Tests

To fully understand the challenge in utilizing the above definitions for CAPA, we have outlined the current diagnostic tools for CAPA, including culture data, PCR, point-of-care tests, and fungal biomarkers. The performance of these tests to diagnose aspergillosis is mostly based on pre-COVID-19 data. However, for patients with CAPA, the performance of these tests may differ, and for some, the performance is still unknown. Specifics are discussed below.

#### 2.2.1. Culture Data

Culture data to diagnose CAPA includes cultures from sputum, endotracheal tube aspirate (ETA), and BAL. Tracheal aspirate and sputum cultures are straightforward to obtain in most critically ill patients who are mechanically ventilated but may represent colonization rather than true infection [31]. From data before COVID-19, sputum cultures have around 50% sensitivity and between 20% and 70% specificity for IPA [32]. Cultures from BAL fluid in intubated patients have been shown to have between 30% and 60% sensitivity and 50% specificity for IPA [33]. Data from BAL culture in CAPA has identified sensitivity to be 53% and specificity to be 100% [34]. Further data on the sensitivity and specificity of culture for the diagnosis of CAPA are unknown [6].

#### 2.2.2. PCR Diagnostics

PCR data to diagnose CAPA can be obtained from the serum, sputum, and BAL. Data before COVID-19 describes the pooled sensitivity and specificity of PCR from serum for IPA as both roughly 80% for a single positive test result and 60% sensitivity and 95% specificity for two consecutive positive test results [35]. From the sputum, sensitivity for IPA is 89% and specificity is 85% [36]. PCR from BAL fluid is more specific than from serum for the diagnosis of IPA, with 88% sensitivity and 95% specificity [37]. One partially prospective and partially retrospective study involving 823 patients using the ECMM/ISHAM definitions found that BAL PCR for the diagnosis of CAPA had 42% sensitivity and 100% specificity [38]. Other data on PCR characteristics in the diagnosis of CAPA are unknown [6].

#### 2.2.3. Point of Care Tests

The lateral flow assay (LFA) for *Aspergillus* can be performed on serum or fluid from BAL. One study found that the LFA from a BAL had a sensitivity of 74% and specificity of 82% for IPA at an index cut-off of 1.5 [39]. Serum LFA has a different performance, with a sensitivity of 49% and specificity of 95% for IPA [40]. In the COVID-19 era, one study examined the sensitivity and specificity of the LFA from BAL fluid based on criteria from EORTC/MSGERC and the AspICU criteria and found that depending on the definitions used, the sensitivity of the LFA ranged from 88% to 94% and the specificity was 81% [41]. Another retrospective multi-center study using the ECMM/ISHAM definitions found that at the 1.0 cutoff using BAL fluid, sensitivity was 52% and specificity was 98% for proven CAPA [42]. Both serum and BAL LFA index are included in the ECCM/ISHAM definitions for probable and possible CAPA.

#### 2.2.4. Fungal Biomarkers

Fungal biomarkers include GM from the serum and BAL fluid and serum Beta-D Glucan (BDG). In the pre-COVID-19 era, the highest performing test in this category is the detection of galactomannan from BAL fluid, with sensitivity greater than 90% [43]. Data from a prospective observational study before COVID-19 showed that two consecutive positive serum BDG tests among patients with proven and probable invasive fungal disease based on the EORTC/MSGERC definitions have a specificity of 86% [44]. One retrospective, single-center study that included 69 ICU patients found that both specificity and sensitivity of serum BDG for CAPA, defined by the ECMM/ISHAM criteria, were 56% [45]. Those authors concluded that serum BDG likely had no role in diagnosing or ruling out CAPA in settings where the prevalence of CAPA is less than 15%. Serum galactomannan has the best performance in patients with hematologic malignancies with classic invasive aspergillosis, as they typically develop angioinvasion leading to antigenemia [46]. However, even then, serum GM sensitivity in patients with IPA is only 30–50% in critically ill patients [47]. Many patients with CAPA lack this classic angioinvasion because of systemic clearance by neutrophils in these non-neutropenic patients [48]. Moreover, the optimal galactomannan index has not been defined [6,7]. Data from a multicenter prospective cohort study in Italy defined CAPA as a patient with COVID-19 requiring ICU-level care who has pulmonary infiltrates plus one of the following: serum GM index > 0.5, BAL GM index ≥ 1.0, positive BAL culture or cavitary lung lesion. The study found that out of 108 ICU patients, 27.7% had probable CAPA, but serum GM was positive in only 1 of those cases [10]. One outlier prospective, a single-center study involving 105 patients found that BAL GM has a sensitivity ranging from 60–80% compared to serum GM sensitivity of 56–80%. [49]. Another study found that the sensitivity of serum galactomannan was <20% [47]. A separate study found that the sensitivity of serum BDG ≥ 80 pg/mL had a sensitivity of 38% and specificity of 85% [34]. Both serum and BAL GM are used in the ECCM/ISHAM criteria to diagnose both probable and possible CAPA. Serum BDG is not included in the ECCM/ISHAM criteria [6].

## 3. Current Data on Antifungal Prophylaxis

Among patients without COVID-19, antifungal prophylaxis is often indicated when there will be prolonged neutropenia, typically in the setting of malignancy, chemotherapy, or other immunosuppression [50]. Medication selection is often between triazole agents, amphotericin B, echinocandins, or combination therapy with voriconazole with an echinocandin, liposomal amphotericin B with an echinocandin, or amphotericin B with a triazole [51]. Posaconazole has been shown to be effective in reducing the incidence of IPA and mortality among neutropenic patients with acute myeloid leukemia and those with graft-versus-host disease after allogeneic hematopoietic stem cell transplant [52,53]. Inhaled amphotericin B has also been proven to be successful in reducing the incidence of IPA in patients who recently underwent lung transplantation and some data suggest the use of inhaled amphotericin B in patients with hematologic malignancies when azole medications are contraindicated [54,55,56,57,58,59]. Inhaled amphotericin is often able to be successfully deposited in the airway and lung parenchyma at adequate levels [60]. Overall the selection of which patients receive prophylaxis and the duration of prophylaxis often vary by institution.

As of the writing of this review, there are no antifungal medications that are listed as clinically indicated for prophylaxis for patients in the ICU. Prior to the COVID-19 pandemic, antifungal prophylaxis has been studied in critically ill patients with influenza. We only know of one randomized open-label trial that compared 7 days of intravenous (IV) posaconazole prophylaxis with no antifungal prophylaxis for patients with respiratory failure due to influenza virus [61]. The authors found no difference in IAPA, but their study was underpowered because most of the patients with IAPA were diagnosed within 48 h of ICU admission, excluding them from the study. The number of IAPA cases in the prophylaxis group was less than in the non-prophylaxis group, but this result was not statistically significant. There was an additional sub-study to understand posaconazole pharmacokinetics for critically ill patients with IAPA and the authors felt that IV posaconazole was appropriate for prophylaxis but not treatment because the posaconazole trough concentration was unable to be reached at sufficient treatment-level doses [62]. Additionally, the prophylaxis group did not see any impact on mortality, duration of respiratory support, or length of stay. However, mortality of early IAPA was up to 53% despite very prompt antifungal treatment.

In patients with COVID-19, there have been few studies investigating the role of antifungal prophylaxis. The most current guidelines and consensus criteria recommend against antifungal prophylaxis for mechanically ventilated COVID-19 patients, but do not discuss the evidence behind this recommendation [6,26]. Previous studies suggested that antifungal prophylaxis was beneficial only when baseline rates of invasive fungal infections were greater than 15% to 30% [63,64,65,66,67,68]. However, it is not clear that we can use the data from the 1990s to the early 2000s to guide the decision on antifungal prophylaxis in the setting of an ongoing pandemic, changes in fungal epidemiology, and the availability of new fungal diagnostic methods. Regarding CAPA prophylaxis specifically, there are no randomized controlled trials to date. There was an attempted randomized controlled trial investigating the use of isavuconazole for the prevention of CAPA but it was terminated early due to challenges with participant enrollment (Isavu-CAPA Trial; ClinicalTrials.gov ID NCT04707703). There is currently an ongoing multi-center case–control study in Europe for IV posaconazole prophylaxis in critically ill patients (POSACOVID Trial; ClinicalTrials.gov ID NCT05065658), but it is unknown when results will become available.

There have been a total of six publications that specifically comment on antifungal prophylaxis for CAPA (Table 1). Two studies investigated posaconazole and four studies investigated inhaled amphotericin prophylaxis. Both studies on posaconazole prophylaxis were prospective. One study measured plasma posaconazole levels after using therapeutic liquid posaconazole suspensions in seven intubated patients via oral-gastric tube [69]. The authors observed very low plasma levels in all patients receiving prophylaxis and recommended against liquid posaconazole prophylaxis for CAPA, but the authors did not specifically comment on whether the patients receiving prophylaxis developed CAPA during their hospitalization. Another study compared patients with and without antifungal prophylaxis, with the majority of those receiving prophylaxis (98%) getting standard dosage IV posaconazole within 48 h of ICU admission [4]. The authors found that out of 132 patients, only 10 were diagnosed with CAPA, and of those, 9 patients did not receive antifungal prophylaxis. However, there was no difference in survival between the two groups.

Among the four studies investigating inhaled amphotericin prophylaxis, there were two prospective and two retrospective studies. The formulation, dose, and duration of amphotericin treatment varied between all studies, and in some studies, this information is not fully available. One small prospective study published out of Belgium reported that since their empiric use of CAPA prophylaxis with 12.5 mg of nebulized liposomal amphotericin B in all ICU patients with COVID-19, they have not experienced any cases of CAPA [70]. However, the authors do not provide specific data regarding timelines and other patient outcomes. In response to an outbreak of CAPA in their hospital, another center implemented prophylaxis for all mechanically ventilated COVID-19 patients with 50 mg of inhaled amphotericin B lipid complex every 48 h. The authors reported that no subsequent patients who received prophylaxis developed CAPA [71]. Reported adverse effects of the prophylaxis included bronchospasm in 8.8% of patients and drug buildup in the expiratory limb’s filter of the ventilator. A retrospective cohort study out of the Netherlands started daily conventional nebulized amphotericin B as antifungal prophylaxis for COVID-19 patients on invasive mechanical ventilation [72]. The study used 10 mg twice daily for 22 days followed by 5 mg four times daily for an unspecified total duration. The authors found that the patients who received prophylaxis had lower incidence of CAPA when compared to the no antifungal group (27% vs. 67%, *p* = 0.047). Again, there was no mortality difference. While the prophylaxis group did not experience any treatment-related adverse events, they were noted to have a longer duration of ICU stay compared to the non-prophylaxis group. The fourth study used twice weekly inhaled liposomal amphotericin B at 12.5 mg for each dose combined with five drops of salbutamol [73]. In their retrospective observational study, the authors found that prophylaxis reduced the incidence of their combined primary outcome of proven or probable CAPA or *Aspergillus* tracheobronchitis as well as the incidence of *Aspergillus* colonization. The authors did not comment on whether mortality outcomes differed [73].

## 4. Our Opinions on Antifungal Prophylaxis

The existing data, though sparse, suggest that antifungal prophylaxis may reduce the incidence of CAPA among mechanically ventilated patients with severe COVID-19. Additionally, prophylaxis may also reduce the incidence of *Aspergillus* colonization, which is theorized to reduce subsequent risks of clinical infection [74]. However, the clinical significance of the limited data needs to be explored. Limited data on mortality is also challenging to interpret given that a clinician electing to give a certain patient antifungal prophylaxis is likely influenced by the severity of that patient’s illness. Additionally, the adverse effects of medications must be weighed against the potential benefits. In the above studies, adverse effects were minimal, but, again, data are limited. Randomized controlled trials are needed to guide clinical decisions. Before these data become available, it may be reasonable to include the following factors when considering antifungal prophylaxis: local prevalence of CAPA, host risk factors, fungal colonization, and history of previous fungal infection.

Before the results of randomized controlled trials become available, the decision to use prophylaxis for CAPA will depend on institutional guidelines and the availability of antifungal agents. With limited data, institutions may need to rely on the local prevalence of CAPA and the likelihood of patients developing the disease. Antifungal prophylaxis has been previously noted to only be beneficial when baseline rates of invasive fungal infections are greater than 15% to 30% [63,65,66,67,68] However, many of these data are from the early 2000s and may not be reflective of current trends during the COVID-19 pandemic. Importantly, the prevalence of CAPA has been shown to vary between hospitals, ranging between 5–30% [3,4,8,12,75,76,77,78] Baseline rates of CAPA also have the potential to wax and wane with different waves of the pandemic. Offering prophylaxis in ICUs with low baseline rates of the disease may yield an overall higher adverse effect-to-benefit ratio than in ICUs with higher baseline rates of disease. Targeted use of prophylaxis using a certain threshold determined by individual hospital centers and ICUs may therefore be useful.

If institutions are electing to use antifungal prophylaxis for CAPA, it is important to consider the agent of choice for prophylaxis, the monitoring of that agent, and the timeline of prophylaxis. The two most studied medications in CAPA prophylaxis are oral or IV posaconazole and inhaled amphotericin. As stated above, antifungal prophylaxis prior to the COVID-19 pandemic has mostly been studied in patients with hematologic malignancies, where the first line includes azole medications [52]. One study found that liquid posaconazole suspension via an oral gastric tube had poor bioavailability in critically ill COVID-19 patients [69]. However, IV posaconazole may be an appropriate option for prophylaxis [62]. Additionally, systemic azoles come with their own risks including hepatotoxicity, drug–drug interactions, QT prolongation, and the challenge of achieving therapeutic levels [79,80]. Nebulized amphotericin B is a good option. However, nebulized drug deposition in ventilated lungs does not have as much supportive data as nebulized prophylaxis in non-ventilated patients. Additionally, nebulized prophylaxis may lead to the clogging of ventilator filters, though this has not been felt to be clinically significant [72]. If we were to select antifungal prophylaxis for our patients, we would elect to pursue inhaled amphotericin B, as this medication is slightly better supported than posaconazole for CAPA prophylaxis specifically. Additionally, many critically ill COVID-19 patients in ICUs have multi-organ failure and are on additional life support, including renal replacement therapy (RRT) and extracorporeal membrane oxygenation (ECMO), which can alter the bioavailability of azole medications. We acknowledge, however, that the use of inhaled amphotericin can potentially cause infection control issues and systemic azoles are typically considered the first line for other antifungal prophylaxis. Selection of prophylaxis is subject to the availability of antifungal agents including their forms in each center as well as local drug resistance profile.

As we write this review on CAPA prophylaxis, it is important to reflect on the changing landscape of both COVID-19 and CAPA. As vaccine availability and outpatient treatment options increase and as COVID-19 variants change, we have seen fewer critically ill patients, and therefore we may not see as many patients with CAPA. This brings into question the relevance of the fungal-after-viral phenomenon and how long it will be problematic for the COVID-19 pandemic. We argue that although the classic hospitalizations of COVID-19 patients may be changing, the fungal-after-viral phenomenon will become even more relevant to certain patient populations, most specifically, the immunocompromised patient population. As we see this shift in many healthcare institutions, it will be even more important to understand the possible benefits of CAPA prophylaxis to protect this vulnerable population. It may be that CAPA prophylaxis becomes common not just in ICUs, but in any non-intubated, immunocompromised patient hospitalized with COVID-19. Early on in the COVID-19 pandemic, one study found that up to 20% of lung transplant recipients developed secondary fungal infections, even those who did not require intubation [81]. There are currently active ongoing studies investigating secondary fungal infections in lung transplant recipients, and institutions may be leaning towards more liberal use of antifungal prophylaxis in this population. Other efforts are being performed to better understand the fungal-after-viral phenomenon in a multitude of immunocompromised patients, including solid organ transplant recipients, bone marrow transplant recipients, patients with malignancies, HIV, and rheumatologic conditions. These will all inform further work on antifungal prophylaxis in COVID-19 patients. Still, randomized controlled trials are warranted to study the safety and efficacy of antifungal prophylaxis for CAPA, including in immunocompromised hosts. Until then, the decision to use antifungal prophylaxis lies in the hands of individual institutions, considering the above information.

## 5. Conclusions

Diagnosis, treatment, and prophylaxis of CAPA are challenging tasks. In this review, we summarized the existing literature on diagnosis, including its difficulties, and reviewed the limited data on prophylaxis for CAPA. Given the difficulties involved in obtaining invasive sampling, improved non-invasive diagnostics that enable screening are needed. Randomized controlled trials are also needed to study the success of antifungal prophylaxis for COVID-19 patients.

## Figures and Tables

**Figure 1 antibiotics-11-01704-f001:**
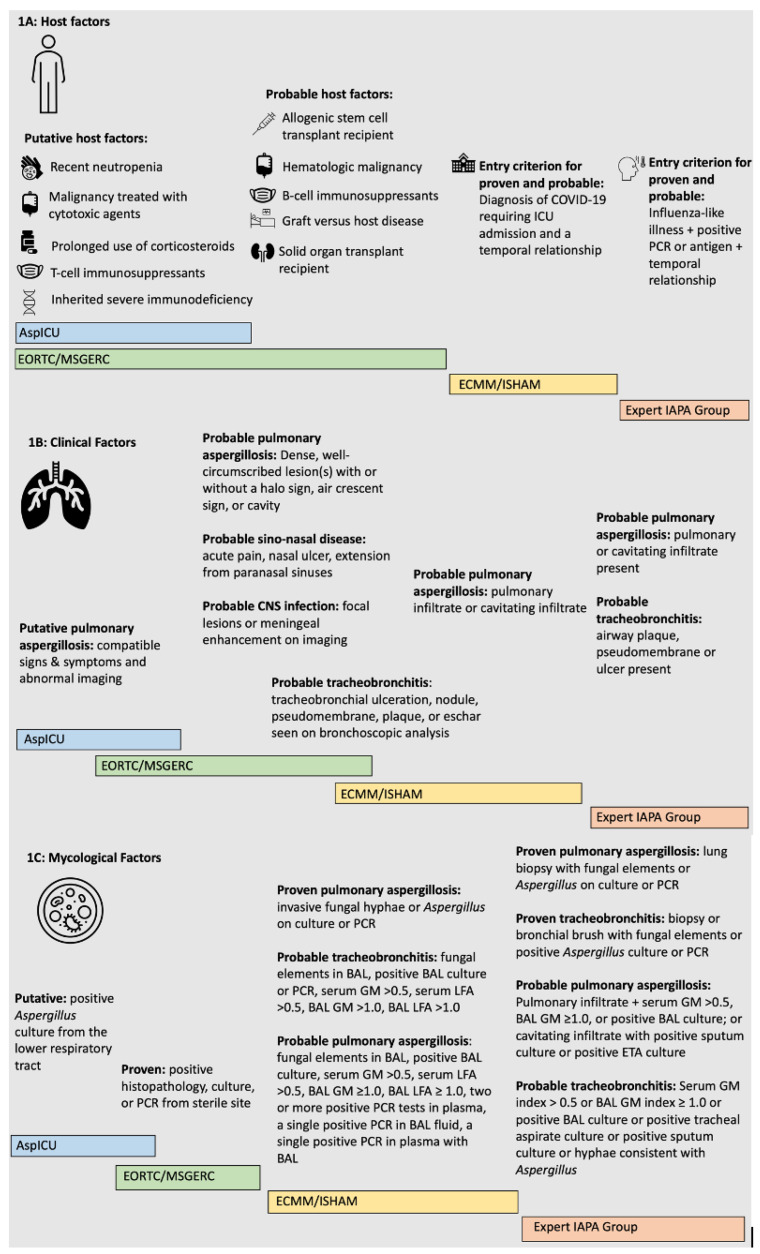
Case definitions of proven and probable CAPA and IAPA (1A: host factors; 1B: clinical factors; 1C mycological factors). CAPA: COVID-19-associated pulmonary aspergillosis; IAPA: influenza-associated pulmonary aspergillosis; EORTC/MSGERC: European Organization for Research and Treatment of Cancer/Invasive Fungal Infections Cooperative Group and the National Institute of Allergy and Infectious Diseases Mycoses Study Group; ECMM/ISHAM: European Confederation for Medical Mycology and the International Society for Human and Animal Mycology; BAL: bronchoalveolar lavage; GM: galactomannan; PCR: polymerase chain reaction.

**Table 1 antibiotics-11-01704-t001:** Summary of existing studies on antifungal prophylaxis for CAPA.

Authors & Country	Study Design	CAPA Definition Used	N	Prophylaxis Used	CAPA Diagnosis (Prophylaxis vs. No Prophylaxis)	Mortality	Overall Summary & Author Recommendations
Mian et al. in the Netherlands	Prospective	Not specified	7	Oral Posaconazole 200 mg suspension three times daily ranging from 8–21 total doses	Not studied	Not studied	-Posaconazole suspension has poor bioavailability -Posaconazole suspension is not recommended to prevent CAPA
Hatzl et al. in Austria	Prospective	ECMM/ISHAM	132	Intravenous Posaconazole (dose, duration, and schedule not specified)	1.4% vs. 17.5%	No difference	-CAPA is associated with poor outcomes -Antifungal prophylaxis can prevent CAPA
Rutsaert et al. in Belgium	Prospective	Not specified	N/A	Nebulized liposomal amphotericin B 12.5 mg (duration and schedule not specified)	Not specified	Not studied	-There should be a low threshold for screening, prophylaxis, and early antifungal treatment for CAPA
Soriano et al. in Spain	Prospective	ECMM/ISHAM	45	Nebulized amphotericin B lipid complex 50 mg every 48 h (total duration not specified)	Not studied	Not studied	-Prophylaxis should be considered to control an outbreak of CAPA
Melchers et al. in the Netherlands	Retrospective	ECMM/ISHAM	39	Nebulized conventional amphotericin B 10 mg twice daily for 22 days, followed by 5 mg four times daily (total duration not specified)	27% vs. 67%	No difference	-Prophylaxis may need to start early during hospitalization -Antifungal prophylaxis can prevent CAPA
Van Ackerbroeck et al. in Belgium	Retrospective	Not specified	78	Nebulized liposomal amphotericin B 12.5 mg with 5 drops of salbutamol twice weekly	9% vs. 61%	Not studied	-Antifungal prophylaxis can reduce the incidence of CAPA and reduce the incidence of aspergillus colonization

CAPA: COVID-19-associated pulmonary aspergillosis; ECMM/ISHAM: The European Confederation for Medical Mycology and the International Society for Human and Animal Mycology; N: number; N/A: not available.

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
