# Peer review of "Diagnosis and Antifungal Prophylaxis for COVID-19 Associated Pulmonary Aspergillosis"

_antibiotics, 2022, doi:10.3390/antibiotics11121704_

Round 1

Reviewer 1 Report

The manuscript entitled" Diagnosis and Antifungal Prophylaxis for COVID-19 Associated Pulmonary Aspergillosis" is focusing on an important subject but the titles of the headings are not clear enough. Also, if a table containing the fungal biomarkers is added, it will be better. Also, I suggest adding a figure describing the antifungal prophylaxis mechanism of action.   

Author Response

Thank you very much for your feedback to improve our manuscript. The data of fungal biomarkers in CAPA diagnosis is extremely limited and we do not think it is suitable for Table. Although we appreciate the reviewer’s suggestion, we do not have enough data on mechanism of antifungal prophylaxis and it would not be appropriate to add a Figure here.  We do want to apologize because we realized our Table 1 was not included in the original document, so we are not sure if you saw this. We have since updated Table 1 to be included in the manuscript document. Additionally, we have reviewed our manuscript to double check that there are no English errors and we feel that it is appropriate. We appreciate your review.

Reviewer 2 Report

Major comments:

The number of studies on CAPA in not enouph. There is just Six studies about CAPA and in this studies mortality after  Antifungal Prophylaxis for CAPA is not studied or is not different.

Author Response

Thank you very much pointing this out. We agree with the reviewer that we need more studies to address this important question. However, we believe that our review will help inform readers regarding current available data. In reality, providers may be forced to make a decision on antifungal prophylaxis with very limited information. We are also hopeful that our work will lead to more research in this topic, which will lead to a collective of more information.

Reviewer 3 Report

The present manuscript is literature based study on to explore the major insights in diagnosis and antifungal prophylaxis for CAPA. the review compilation nicely presented the difficulties involved in obtaining invasive sampling, improved non-invasive diagnostics that enable screening and Randomized controlled trials are needed to study the success of antifungal prophylaxis for COVID-19 patients. This manuscript is well in journal's scope and quite informative.

Thanks.

Author Response

Thank you very much for your kind feedback. We appreciate your comments.

Round 2

Reviewer 2 Report

Dear Authors

Surely your study will be useful for those who work in this field.